# Neural approximation of Wasserstein distance via a universal architecture for symmetric and factorwise group invariant functions

**Samantha Chen**
Department of Computer Science Engineering
University of California - San Diego
`sac003@ucsd.edu`

**Yusu Wang**
Halıcıoğlu Data Science Institute
University of California - San Diego
`yusuwang@ucsd.edu`

## Abstract

Learning distance functions between complex objects, such as the Wasserstein distance to compare point sets, is a common goal in machine learning applications. However, functions on such complex objects (e.g., point sets and graphs) are often required to be invariant to a wide variety of group actions e.g. permutation or rigid transformation. Therefore, continuous and symmetric *product* functions (such as distance functions) on such complex objects must also be invariant to the *product* of such group actions. We call these functions *symmetric and factor-wise group invariant functions* (or *SFGI functions* in short). In this paper, we first present a general neural network architecture for approximating SFGI functions. The main contribution of this paper combines this general neural network with a sketching idea to develop a specific and efficient neural network which can approximate the $p$-th Wasserstein distance between point sets. Very importantly, the required model complexity is *independent* of the sizes of input point sets. On the theoretical front, to the best of our knowledge, this is the first result showing that there exists a neural network with the capacity to approximate Wasserstein distance with bounded model complexity. Our work provides an interesting integration of sketching ideas for geometric problems with universal approximation of symmetric functions. On the empirical front, we present a range of results showing that our newly proposed neural network architecture performs comparatively or better than other models (including a SOTA Siamese Autoencoder based approach). In particular, our neural network generalizes significantly better and trains much faster than the SOTA Siamese AE. Finally, this line of investigation could be useful in exploring effective neural network design for solving a broad range of geometric optimization problems (e.g., $k$-means in a metric space).

## 1   Introduction

Recently, significant interest in geometric deep learning has led to a focus on neural network architectures which learn functions on complex objects such point clouds [34, 22] and graphs [23]. Advancements in the development of neural networks for complex objects has led to progress in a variety of applications from 3D image segmentation [28] to drug discovery [3, 13]. One challenge in learning functions over such complex objects is that the desired functions are often required to be invariant to certain group actions. For instance, functions on point clouds are often permutation invariant with respect to the ordering of individual points. Indeed, developing permutation invariant or equivariant neural network architectures, as well as understanding their universal approximation properties, has attracted significant attention in the past few years; see e.g., [22, 34, 19, 20, 25, 16, 33, 29, 30, 4]

37th Conference on Neural Information Processing Systems (NeurIPS 2023).

However, in many geometric or graph optimization problems, our input goes beyond a single complex object, but multiple complex objects. For example, the $p$-Wasserstein distance $W_p(X, Y)$ between two point sets $X$ and $Y$ sampled from some metric space (e.g., $\mathbb{R}^d$) is a function over pairs of point sets. To give another example, the 1-median of the collection of $k$ point sets $P_1, \ldots, P_k$ in $\mathbb{R}^d$ can be viewed as a function over $k$ point sets.

A natural way to model such functions is to use product space. In particular, let $\mathcal{X}$ denote the space of finite point sets from a bounded region in $\mathbb{R}^d$. Then the $p$-Wasserstein distance can be viewed as a function $W_p : \mathcal{X} \times \mathcal{X} \to \mathbb{R}$. Similarly, 1-median for $k$ point sets can be modeled by a function from the product of $k$ copies of $\mathcal{X}$ to $\mathbb{R}$. Such functions are not only invariant to permutations of the factors of the product space (i.e. $W_p(X, Y) = W_p(Y, X)$) but are also invariant or equivariant with respect to certain group actions for each factor. For $p$-Wasserstein distance, $W_p$ is invariant to permutations of the ordering of points within both $X$ and $Y$. This motivates us to extend the setting of learning continuous group invariant functions to learning continuous functions over *product* spaces which are both invariant to some product of group actions and symmetric. More precisely, we consider a type of function which we denote as an SFGI product functions.

**Definition 1.1 (SFGI product function)** *Given compact metric spaces $(\mathcal{X}_i, \mathrm{d}_{\mathcal{X}_i})$ where $i \in [k]$, we define symmetric and factor-wise group invariant (SFGI) product functions as $f : \mathcal{X}_1 \times \mathcal{X}_2 \times \cdots \mathcal{X}_k \to \mathbb{R}$ where $f$ is (1) symmetric over the $k$ factors, and (2) invariant to the group action $G_1 \times G_2 \times \cdots \times G_k$ for some group $G_i$ acting on $\mathcal{X}_i$, for each $i \in [1, k]$.*

**Contributions.** SFGI product functions can represent a wide array of geometric matching problems including computing the Wasserstein or Hausdorff distance between point sets. In this paper, we first provide a general framework for approximating SFGI product functions in Section 3.1. Our primary contribution, described in Section 3.2, is the integration of this general framework with a sketching idea in order to develop an efficient and specific SFGI neural network which can approximate the $p$-Wasserstein distance between point sets (sampled from a compact set in a nice metric space, such as the fixed-dimensional Euclidean space). Most importantly, the complexity of our neural network (i.e., number of parameters needed) is *independent* of the maximum size of the input point sets, and depends on only the additive approximation error. To the best of our knowledge, this is the first architecture which provably achieves this property. This is in contrast to many existing universal approximation results on graphs or sets (e.g., for DeepSet) where the network sizes depend on the size of each graph or point set in order to achieve universality [20, 30, 4]. We also provide a range of experimental results in Section 4 showing the utility of our neural network architecture for approximating Wasserstein distances. We compare our network with both a SOTA Siamese autoencoder [15], a natural Siamese DeepSets network, and the standard Sinkhorn approximation of Wasserstein distance. Our results show that our universal neural network architecture produces Wasserstein approximations which are better than the Siamese DeepSets network, comparable to the SOTA Siamese autoencoder and generalize much better than both to input point sets of sizes which are unseen at training time. Furthermore, we show that our approximation (at inference time) is much faster than the standard Sinkhorn approximation of the $p$-Wasserstein distance at similar error threshholds. Moreover, our neural network trains much faster than the SOTA Siamese autoencoder. Overall, our network is able to achieve **equally accurate or better** Wasserstein approximations which **generalize better** to point sets of unseen size as compared to SOTA while **significantly reducing** training time. In Appendix B, we provide discussion of issues with other natural choices of neural network architectures one might use for estimating Wasserstein distances, including one directly based on Siamese networks, which are often used for metric learning.

All missing proofs / details can be found in the **Supplementary materials**.

**Related work.** Efficient approximations of Wasserstein distance via neural networks are an active area of research. One approach is to use input convex neural networks to approximate the 2-Wasserstein distance[18, 27]. However, for this approach, training is done *per* pair of inputs and is restricted to the 2-Wasserstein distance which makes it unsuitable for a general neural network approximation of $p$-Wasserstein distances between discrete distributions. This neural approximation method contrasts with our goal: a general neural network that can approximate the $p$-Wasserstein distance between any two point sets in a compact metric space to within $\epsilon$-accuracy. Siamese networks are another approach for popular approach for learning Wasserstein distances. Typically, a Siamese network is composed of a single neural network which maps two input instances to Euclidean space.

The output of the network is represented then by $\ell_p$-norm between the output embeddings. In [5], the authors utilize a Siamese autoencoder which takes two histograms (images) as input. For their architecture, a single encoder network is utilized to map each histogram to an embedding in Euclidean space, while a decoder network maps each Euclidean embedding back to an output histogram. The Kullback-Liebler (KL) divergence between original histogram and the output histogram (i.e. reconstruction loss) is used during training to regularize the embeddings and the final estimate of 2-Wasserstein distance is the $\ell_2$ norm between the embeddings. The idea of learning Wasserstein distances via Siamese autoencoders was extended in [15] to point cloud data with the Wasserstein point cloud embedding network (WPCE) where the original KL reconstruction loss was replaced with a differentiable Wasserstein approximation between the original point set and a fixed-size output point set from the decoder network. In our subsequent experiments, we show that our neural network trains much more efficiently and generalizes much better than WPCE to point sets of unseen size.

Moreover, the concept of group invariant networks was previously investigated in several works, including [34, 22, 19, 17, 9]. For instance, DeepSets [34] and PointNet [19] are two popular permutation invariant neural network which were shown to be universal with respect to set functions. In addition to group invariance, there have also been efforts to explore the notion of invariance with respect to combinations of groups, such as invariance to both SE(3) and permutation group [11, 21] or combining basis invariance with permutation invariance [17]. Our work differs from previous work in that we address a universal neural network which is invariant with respect to a *specific* combination of permutation groups that corresponds to an SFGI function on point sets. In general, we can view this as a subgroup of the permutation group - encoding the invariance of each individual point set with the symmetric requirement of the product function corresponds to a specific subgroup of the permutation group. Thus, previous results regarding permutation invariant architectures such as DeepSets, PointNet or combinations of group actions (such as [11]) do not address our setting of SFGI functions or $p$-Wasserstein distances.

## 2   Preliminaries

We will begin with basic background on groups, universal approximation, and Wasserstein distances.

**Groups and group actions**   A group $G$ is an algebraic structure that consists of a set of elements and a binary operation that satisfies a specific set of axioms: (1) the associative property, (2) the existence of an identity element, and (3) existence of inverses for each element in the set. Given a metric space $(\mathcal{X}, d_{\mathcal{X}})$, the action of the group $G$ on $\mathcal{X}$ is a function $\alpha : G \times \mathcal{X} \to \mathcal{X}$ that transforms the elements of $\mathcal{X}$ for each element $\pi \in G$. For each element $\pi \in G$, we will write $\pi \cdot x$ to denote the action of a group element $\pi$ on $x$ instead of $\alpha(\pi, x)$. For example, if $G$ is the permutation group over $[N] := \{1, 2, \ldots, N\}$, and $\mathcal{X} = \mathbb{R}^N$, then for any $\pi \in G$, $\pi \cdot x$ represents the permutation of elements in $x \in \mathbb{R}^N$ via $\pi$ i.e. given $x = (x_1, x_2, \ldots, x_N)$, $\pi \cdot x = (x_{\pi(1)}, x_{\pi(2)} \ldots, x_{\pi(N)})$. A function $f : \mathcal{X} \to \mathcal{Y}$ is *G-invariant* if for any $X \in \mathcal{X}$ and any $\pi \in G$, we have that $f(X) = f(\pi \cdot X)$.

**Universal Approximation.**   Let $\mathcal{C}(\mathcal{X}, \mathbb{R})$ denote the set of continuous functions from a metric space $(\mathcal{X}, d_{\mathcal{X}})$ to $\mathbb{R}$. Given two families of functions $\mathcal{F}_1$ and $\mathcal{F}_2$ where $\mathcal{F}_1 \subseteq \mathcal{F}_2$ and $\mathcal{F}_1, \mathcal{F}_2 \subseteq \mathcal{C}(\mathcal{X}, \mathbb{R})$, we say that $\mathcal{F}_1$ *universally approximates* $\mathcal{F}_2$ if for any $\epsilon > 0$ and any $f \in \mathcal{F}_2$, there is a $g \in \mathcal{F}_1$ such that $\|g - f\|_{\infty} < \epsilon$. Different norms on the space of functions can be used, but we will use $L_{\infty}$ norm in this paper, which intuitively leads to additive pointwise-error over the domain of these functions. The classic universal approximation theorem for multilayer perceptrons (MLPs) [7] states that a feedforward neural network with a single hidden layer, using certain activation functions, can approximate any continuous function to within an arbitrary *additive $\epsilon$-error.*

**Permutation invariant neural networks for point cloud data**   One of the most popular permutation invariant neural network models is the DeepSets model defined in [34]. DeepSets is designed to handle unordered input point sets by first applying a neural network to each individual element, then using sum-pooling to generate an embedding for the input data set, and finally, applying a final neural network architecture to the ouput embedding. Formally, suppose we are given a finite *multiset* $S = \{x : x \in \mathbb{R}^d\}$ (meaning that an element can appear multiple times, and the number of times an

element occurs in $S$ is called its *multiplicity*). The DeepSets model is defined as

$$\mathcal{N}_{\text{DeepSet}}(S) = g_{\theta_2}\Big(\sum_{x \in S} h_{\theta_1}(x)\Big)$$

where $h_{\theta_1}$ and $g_{\theta_2}$ are neural network architectures. DeepSets can handle input point sets of variable sizes. It was also shown to be universal with respect to continuous *multiset functions*.

**Theorem 2.1 ([34, 1, 12])** *Assume the elements are from a compact set in $\mathbb{R}^k$, and the input multiset size is fixed as $N$. Let $t = 2kN + 1$. Then any continuous multiset function, represented as $f : \mathbb{R}^{k \times N} \to \mathbb{R}$ which is invariant with respect to permutations of the columns, can be approximated arbitrarily close in the form of $\rho\Big(\sum_{x \in X} \phi(x)\Big)$, for continuous transformations $\phi : \mathbb{R}^k \to \mathbb{R}^t$ and $\rho : \mathbb{R}^t \to \mathbb{R}$.*

While universality for the case when $k = 1$ was shown using symmetric polynomials, the case for $k > 1$ in fact is quite subtle and the proof in [34] misses key details. For completeness, we provide a full proof in Appendix E.1 for when the output dimension of $\phi$ is $t = \binom{k+N}{k}$. It was recently shown in [12, 1] that the output dimension of $\phi$ can be reduced to $2kN + 1$, which is the dimension of $t$ which we use in Theorem 2.1 and subsequent theorems. In both the cases where the output dimension of $\phi$ is $t = \binom{k+N}{k}$ or $t = 2kN + 1$, Theorem 2.1 implies that if we want to achieve universality, the required network size depends on input point cloud size.

**Wasserstein distances and approximations.** Here we will introduce Wasserstein distance for discrete measures. Let $(X, \mathrm{d}_X)$ be a metric space. For two weighted point sets $P = \{(x_i, w_i) : x_i \in X, \sum_{w_i} = 1, i \in [n]\}$ and $Q = \{(x_i', w_i') : x_i' \in X, \sum_{w_i} = 1, i \in [m]\}$, we define the Wasserstein distance between $P$ and $Q$ as

$$\mathrm{W}_p(P, Q) = \min_{\Pi \in \mathbb{R}_+^{n \times m}} \Big\{ \big( \langle \Pi, D^p \rangle \big)^{1/p} : \Pi \mathbf{1} = [w_1, \dots, w_n], \Pi^T \mathbf{1} = [w_1', \dots, w_m'] \Big\}$$

where $D \in \mathbb{R}_+^{n \times m}$ is the distance matrix with $D_{i,j} = \mathrm{d}_X(x_i, x_j')$. One can think of these weighted point sets as discrete probability distributions in $(X, \mathrm{d}_X)$. When $p = 1$, $\mathrm{W}_1$ is also commonly known as the Earth Mover's distance (EMD). Additionally, note that when $p = \infty$, $\mathrm{W}_p$ is the same as the Hausdorff distance between points in $P$ and $Q$ with non-zero weight. Computing Wasserstein distances amounts to solving a linear programming problem, which takes $O(N^3 \log N)$ (where $N = \max\{n, m\}$) time. There have been a number of methods for fast approximations of Wasserstein distances, including multi-scale and hierarchical solvers [24], and $L_1$ embeddings via quadtree algorithms [2, 14]. In particular, entropic regularization of Wasserstein distance [6], also known as the Sinkhorn distance, is often used as the standard Wasserstein distance approximation for learning tasks. Unlike Wasserstein distance, the Sinkhorn approximation is differentiable and can be computed in approximately $O(n^2)$ time. The computation time is governed by a regularization parameter $\epsilon$. As $\epsilon$ approaches zero, the Sinkhorn distance approaches the true Wasserstein distance.

## 3 Learning functions between point sets

We will first present a general framework for approximating SFGI-functions and then show how this framework along with geometric sketches of our input data enables us to define neural networks which can approximate $p$-Wasserstein distances with complexity independent of input data size.

### 3.1 A general framework for functions on product spaces

One of the key ingredients in our approach is the introduction of what we call a *sketch* of input data to an Euclidean space whose dimension is independent of the size of the input data.

**Definition 3.1 (Sketch)** *Let $\delta > 0$, $a \in \mathbb{N}^+$, and $G$ be a group which acts on $\mathcal{X}$. A $(\delta, a, G)$-sketch of a metric space $(\mathcal{X}, \mathrm{d}_\mathcal{X})$ consists of a $G$-invariant continuous encoding function $h : \mathcal{X} \to \mathbb{R}^a$ and a continuous decoding function $g : \mathbb{R}^a \to \mathcal{X}$ such that $\mathrm{d}_\mathcal{X}(g \circ h(S), S) < \delta$.*

Now let $(\mathcal{X}_1, \mathrm{d}_{\mathcal{X}_1}), \ldots, (\mathcal{X}_m, \mathrm{d}_{\mathcal{X}_m})$ be compact metric spaces. The product space $\mathcal{X}_1 \times \cdots \times \mathcal{X}_m$ is still a metric space equipped with the following natural metric induced from metrics of each factor:

$$\mathrm{d}_{\mathcal{X}_1 \times \cdots \times \mathcal{X}_m}((A_1, \ldots, A_m), (A_1', \ldots, A_m')) = \mathrm{d}_{\mathcal{X}_1}(A_1, A_1') + \cdots + \mathrm{d}_{\mathcal{X}_m}(A_m, A_m').$$

Suppose $G_i$ is a group acting on $\mathcal{X}_i$, for each $i \in [m]$. In the following result, instead of SFGI product functions, we first consider the more general case of *factor-wise group invariant functions*, namely functions $f : \mathcal{X}_1 \times \cdots \times \mathcal{X}_m \to \mathbb{R}$ such that $f$ is uniformly continuous and invariant to the group action $G_1 \times \cdots \times G_m$.

**Lemma 3.2** *Suppose $f : \mathcal{X}_1 \times \cdots \times \mathcal{X}_m \to \mathbb{R}$ is uniformly continuous and invariant to $G_1 \times \cdots \times G_m$. Additionally, assume that for any $\delta > 0$, $(\mathcal{X}_i, \mathrm{d}_{\mathcal{X}_i})$ has a $(\delta, a_i, G_i)$-sketch where $a_i$ may depend on $\delta$. Then for any $\epsilon > 0$, there is a continuous $G_i$-invariant functions $\phi_i : \mathcal{X}_i \to \mathbb{R}^{a_i}$ for all $i \in [k]$ and a continuous function $\rho : \mathbb{R}^{a_1} \times \cdots \times \mathbb{R}^{a_m} \to \mathbb{R}$ such that*

$$|f(A_1, A_2, \ldots, A_m) - \rho(\phi_1(A_1), \phi_2(A_2), \ldots, \phi_k(A_m))| < \epsilon$$

*for any $(A_1, \ldots, A_m) \in \mathcal{X}_1 \times \cdots \times \mathcal{X}_m$. Furthermore, if $\mathcal{X}_1 = \ldots \mathcal{X}_2 = \cdots = \mathcal{X}_m$, then we can choose $\phi_1 = \phi_2 = \ldots \phi_m$.*

Note that a recent result from [17] shows that a continuous factor-wise group invariant function $f : \mathcal{X}_1 \times \cdots \mathcal{X}_m \to \mathbb{R}$ can be **represented** (not approximated) by the form $f(v_1, \ldots, v_m) = \rho(\phi_1(v_1), \ldots, \phi_k(v_m))$ if *there exists a topological embedding from $\mathcal{X}_i/G_i$ to Euclidean space*. The condition that each quotient $\mathcal{X}_i/G_i$ has a topological embedding in fixed dimensional Euclidean space is strong. A topological embedding requires injectivity, while in a sketch, one can collapse input objects as long as after decoding, we obtain an approximated object which is close to the input. Our result can be viewed as a relaxation of their result by allowing our space to have an approximate fixed-dimensional embedding (i.e., our $(\delta, a, G)$-**sketch**).

We often consider the case where $\mathcal{X} = \mathcal{X}_1 = \cdots = \mathcal{X}_m$ i.e. $f : \mathcal{X} \times \cdots \mathcal{X} \to \mathbb{R}$ where $G$ is a group acting on the factor $\mathcal{X}$. Oftentimes, we require the function to not only be invariant to the actions of a group $G$ on each individual $\mathcal{X}$ but also *symmetric* with respect to the ordering of the input. By this, we mean $f(A_1, \ldots, A_m) = f(A_{\pi(1)}, \ldots, A_{\pi(m)})$ where $\pi$ is a permutation on $[m]$. In other words, we now consider the *SFGI product function $f$* as introduced in Definition 1.1. The extra symmetry requirement adds more constraints to the form of $f$. We show that the set of uniformly continuous SFGI product function can be universally approximated by product function with an even simpler form than Lemma 3.2 as stated in the theorem below.

**Lemma 3.3** *Assume the same setup as Lemma 3.2 with $\mathcal{X} = \mathcal{X}_1 = \cdots = \mathcal{X}_m$ and $G = G_1 = \cdots = G_m$. Assume that $\mathcal{X}$ has a $(\delta, a, G)$-sketch. Additionally, suppose $f$ is symmetric; hence $f$ is a SFGI function. Let $t = 2am + 1$. Then for any $\epsilon > 0$, there is a continuous $G$-invariant function $\phi : \mathcal{X} \to \mathbb{R}^t$ and a continuous function $\rho : \mathbb{R}^t \to \mathbb{R}$ such that*

$$\left| f(A_1, \ldots, A_m) - \rho\left( \sum_{i=1}^m \phi(A_i) \right) \right| < \epsilon$$

Now suppose we want approximate an SFGI product function, $f$, with a neural network. Lemma 3.3 implies that we can approximate $\phi$ with any universal $G$-invariant neural network which embeds our original space $\mathcal{X}$ to some Euclidean space $\mathbb{R}^a$. Then the outer architecture $\rho$ can be any universal architecture (e.g. MLP). Finding a universal $G$-invariant neural network to realize $\phi$ over a single factor space $\mathcal{X}$ is in general much easier than finding a SFGI neural network, and as we discussed at the end of Section 1, we already know how to achieve this for several settings. We will show how this idea is at work for approximating SFGI functions between point sets in the next subsection.

## 3.2 Universal neural networks for functions between point sets

We are interested in learning symmetric functions between point sets (i.e. any $p$-Wasserstein distance) which are factor-wise *permutation* invariant. In this section, we will show that we can find a $(\delta, a, G)$-sketch for the space of weighted point sets. This allows us to combine Lemma 3.3 with DeepSets to define a set of neural networks which can approximate $p$-th Wasserstein distances to arbitrary accuracy. Furthermore, the encoding and decoding functions can be approximated with neural networks where their model complexity is *independent* of input point set size. Thus, the resulting neural network used to approximate Wasserstein distance also has **bounded model complexity**.

**Set up.** Given some metric space $(\Omega, d_\Omega)$, let $\mathcal{X}$ be the set of *weighted* point sets with at most $N$ elements. In other words, each $S \in \mathcal{X}$ has the form $S = \{(x_i, w_i) : w_i \in \mathbb{R}_+, \sum_i w_i = 1, x_i \in \Omega\}$ and $|S| \leq N$. One can also consider $\mathcal{X}$ to be the set of weighted Dirac measures over $\Omega$. For simplicity, we also sometimes use $S$ to refer to just the set of points $\{x_i\}$ contained within it. We will consider the metric over $\mathcal{X}$ to be the $p$-th Wasserstein distance, $W_p$. We refer to the metric space of weighted point sets over $\Omega$ as $(\mathcal{X}, W_p)$.

First, we will show that given a $\delta$, there is a $(\delta, a, G)$-sketch of $\mathcal{X}$ with respect to $W_p$. The embedding dimension $a$ depends on the so-called *covering number* of the metric space $(\Omega, d_\Omega)$ from which points are sampled. Given a compact metric space $(\Omega, d_\Omega)$, the *covering number* $\nu_\Omega(r)$ *w.r.t. radius* $r$ is the minimal number of radius $r$ balls needed to cover $\Omega$. As a simple example, consider $\Omega = [-\Delta, \Delta] \subseteq \mathbb{R}$. Given any $r$, we can cover $X$ with $\frac{2\Delta}{r}$ intervals so $\nu_\Omega(r) \leq \frac{2\Delta}{r}$. The collection of the center of a set of $r$-balls that cover $\Omega$ an $r$-*net of* $\Omega$. For a compact set $\Omega \subset \mathbb{R}^d$ with diameter $D$, its covering number $\nu_\Omega(r)$ is a constant depending on $D$, $r$ and $d$ only.

**Theorem 3.4** *Set* $d_\mathcal{X}$ *to be* $W_p$ *for* $1 \leq p < \infty$. *Let* $G$ *be the permutation group. For any* $\delta > 0$, *let* $\delta_0 = \frac{1}{2} \sqrt[p]{\delta/2}$ *and* $a = \nu_\Omega(\delta_0)$ *be the covering number w.r.t. radius* $\delta_0$. *Then there is a* $(\delta, a, G)$-*sketch of* $\mathcal{X}$ *with respect to* $W_p$. *Furthermore, the encoding function* $\mathbf{h} : \mathcal{X} \to \mathbb{R}^a$ *can be expressed as the following where* $h : \Omega \to \mathbb{R}^a$ *is continuous:*

$$\mathbf{h}(S) = \sum_{x \in S} h(x). \tag{1}$$

*Proof:* Let $\delta > 0$ and let $S \in \mathcal{X}$ be $S = \{(x_i, w_i) : \sum w_i = 1, x_i \in \Omega\}$ and $|S| \leq N$. Given $\delta_0 = \frac{1}{2} \sqrt[p]{\delta/2}$ and $a = \nu_\Omega(\delta_0)$, we know $\Omega$ has a $\delta_0$-net, $C$, and we denote the elements of $C$ as $\{y_1, \ldots, y_a\}$. In other words, for any $x \in \Omega$, there is a $y_i \in Cd$ such that $d_\Omega(x, y_i) < \delta_0$.

First, we will define an encoding function $\rho : \mathcal{X} \to \mathbb{R}^a$. For each $y_i$, we will use a soft indicator function $e^{-bd_\Omega(x, B_{\delta_0}(y_i))}$ and set the constant $b$ so that $e^{-bd_\Omega(x, B_{\delta_0}(y_i))}$ is "sufficiently" small if $d_\Omega(x, B_{\delta_0}(y_i)) > \delta_0$. More formally, we know that $\lim_{b \to \infty} e^{-b\delta_0} = 0$ so there is $\beta \in \mathbb{R}$ such that for all $b > \beta$, $e^{-b\delta_0} < \frac{\delta_0^p}{d_{max}^p \cdot a}$. Set $b_0$ to be such that $b_0 > \beta$. Let $h_i(x) = e^{-b_0 d_\Omega(x, B_{\delta_0}(y_i))}$ for each $i \in [a]$. For a given $x \in \Omega$, we compute $h : \Omega \to \mathbb{R}^a$ as

$$h(x) = [h_1(x), \ldots, h_a(x)]$$

Then we define the encoding function $\mathbf{h} : \mathcal{X} \to \mathbb{R}^a$ as

$$\mathbf{h}(S) = \sum_{i=1}^{n} w_i \frac{h(x_i)}{\|h(x_i)\|_1}$$

Note that $\|\mathbf{h}(S)\|_1 = 1$ and $\mathbf{h}$ is continuous since Wasserstein distances metrize weak convergence. Additionally, since $d_\Omega(x, B_{\delta_0}(y_i))$ is the distance from $x$ to the $\delta_0$-ball around $y_i$, we are guaranteed to have one $j$ where $h_j(x_i) = 1$ so $\|h(x_i)\|_1 > 1$.

Now, we define a decoding function $g : \mathbb{R}^a \to \mathcal{X}$ as $g(v) = \{(y_i, \frac{v_i}{\|v\|_1}) : i \in [a]\}$. In order to show that $g$ and $\mathbf{h}$ yields a valid $(\delta, a, G)$-sketch of $\mathcal{X}$, we must show that $g \circ \mathbf{h}(S)$ is sufficiently close to $S$ under the $W_p$ distance. First, we know that

$$W_p^p(g \circ \mathbf{h}(S), S) \leq \sum_{i=1}^{n} \sum_{j=1}^{a} w_1 \frac{h_j(x_i)}{\|h(x_i)\|_1} d(x_i, y_j)^p.$$

Let $d_{max}$ be the diameter of $\Omega$. For a given $x_i$, we can partition $\{h_1(x_i), \ldots, h_a(x_i)\}$ into those where $h_j(x_i) \geq \frac{\delta_0^p}{d_{max}^p \cdot a}$ and those where $h_j(x_i) < \frac{\delta_0^p}{d_{max}^p \cdot a}$ i.e. $\{h_{j_1}(x_i), \ldots, h_{j_k}(x_i)\}$ and $\{h_{j_{k+1}}(x_i), \ldots, h_{j_a}(x_i)\}$ respectively. If $h_j(x) \geq \frac{\delta_0^p}{d_{max}^p \cdot a}$, then

$$e^{-b_0 d_\Omega(x, B_{\delta_0}(y_i))} \geq \frac{\delta_0^p}{d_{max}^p \cdot a} > e^{-b_0 \delta_0}$$

so $d_\Omega(x, B_{\delta_0}(y_i)) < \delta_0$. Then

$$W_p^p(g \circ \mathbf{h}(S), S) \leq \sum_{i=1}^n \sum_{j=1}^m w_i \frac{h_j(x_i)}{\|h(x_i)\|_1} d_\Omega(x_i, y_j)^p.$$

$$< \sum_{i=1}^n w_i \Big( \sum_{\ell=1}^k \frac{h_{j_\ell}(x_i)}{\|h(x_i)\|_1} (2\delta_0)^p + \sum_{\ell=k+1}^a \frac{\delta_0^p}{d_{max}^p \cdot a} d_{max}^p \Big) \text{ since } d_\Omega(x_i, y_j) \leq d_{max}$$

$$\leq \sum_{i=1}^n w_i (2^p \delta_0^p + \delta_0^p) \leq 2^p \Big( \sqrt[p]{\delta/2} \cdot \frac{1}{2} \Big)^p + \frac{1}{2^p} \cdot \frac{\delta}{2} < \frac{\delta}{2} + \frac{\delta}{2} = \delta$$

Thus, the encoding function $\mathbf{h}$ and the decoding function $g$ make up a $(\delta, a, G)$-sketch. ∎

Note that the sketch outlined in Theorem 3.4 is a smooth version of a one-hot encoding. With Theorem 3.4 and Lemma 3.3, we will now give an explicit formulation of an $\epsilon$-approximation of $f$ via sum-pooling of continuous functions.

**Corollary 3.5** *Let $\epsilon > 0$, $(\Omega, d_\Omega)$ be a compact metric space and let $\mathcal{X}$ be the space of weighted point sets equipped with the p-Wasserstein, $W_p$. Suppose for any $\delta$, $(\Omega, d_\Omega)$ has covering number $a(\delta)$. Then given a function $f : \mathcal{X} \times \mathcal{X} \to \mathbb{R}$ that is uniformly continuous and permutation invariant, there is continuous functions $h : \Omega \to \mathbb{R}^{a(\delta)}$, $\phi : \mathbb{R}^{a(\delta)} \to \mathbb{R}^{a'}$, and $\rho : \mathbb{R}^{a'} \to \mathbb{R}$, such that for any $A, B \in \mathcal{X}$*

$$\left| f(A, B) - \rho \Big( \phi \Big( \sum_{(x, w_x) \in A} w_x h(x) \Big) + \phi \Big( \sum_{(x, w_x) \in B} w_x h(x) \Big) \Big) \right| < \epsilon$$

*where $h$, $\phi$ and $\rho$ are all continuous and $a' = 4 \cdot a(\delta) + 1$.*

Due to Eqn. (1), instead of considering the function $\mathbf{h}$ which takes a set of points $S \in \mathcal{X}$ as input, we now only need to model the function $h : \Omega \to \mathbb{R}^a$, which takes **a single point** $x \in S$ as input. For simplicity, assume that the input metric space $(\Omega, d_\Omega)$ is a compact set in some Euclidean space $\mathbb{R}^d$. Note that in contrast to Lemma 3.3, each $h$, $\phi$ and $\rho$ is simply a continuous function, and there is no further group invariance requirement. Furthermore, all the dimensions of the domain and range of these functions are bounded values that depend only on the covering number of $\Omega$, the target additive error $\epsilon$, and independent to the maximum size $N$ of input points. We can use multilayer perceptrons (MLPs) $h_{\theta_1}$, $\phi_{\theta_2}$, and $\rho_{\theta_3}$ in place of $h$, $\phi$ and $\rho$ to approximate the desired function. Formally, we define the following family of neural networks:

$$\mathcal{N}_{\text{ProductNet}}(A, B) = \rho_{\theta_3} \Big( \phi_{\theta_2} \Big( \sum_{(x, w_x) \in A} w_x h_{\theta_1}(x) \Big) + \phi_{\theta_2} \Big( \sum_{(x, w_x) \in B} w_x h_{\theta_1}(x) \Big) \Big). \quad (2)$$

In practice, we consider the input to the the neural network $h_{\theta_1}$ to be a point $x \in \Omega$ along with its weight $w_x$. As per the discussions above, functions represented by $\mathcal{N}_{\text{ProductNet}}$ can universally approximate SFGI product functions on the space of point sets. See Figure 1 for an illustration of our universal architecture for approximating product functions on point sets. As $p$-Wasserstein distances, $W_p : \mathcal{X} \times \mathcal{X} \to \mathbb{R}$, are uniformly continuous with respect to the underlying metric $W_p$, we can apply our framework for the problem of approximating $p$-Wasserstein.

Importantly, the number of parameters in $\mathcal{N}_{\text{ProductNet}}$ does not depend on the maximum size of the point set but rather only on the $\epsilon$ additive error and by extension, the covering number of the original metric space. This is because the encoding function for our sketch is defined as the summation of single points into $\mathbb{R}^a$ where $a$ is independent of the size of the input set. Contrast this result with latent dimension in the universality statement of DeepSets (cf. Theorem 2.1), which is dependent on the input point set size. Note that in general, the model complexities of the MLPs $\rho_{\theta_3}$, $\phi_{\theta_2}$, and $h_{\theta_1}$ depend on the dimensions of the domain and co-domain of each function they approximate ($\rho$, $\phi$ and $h$) and the desired approximation error $\epsilon$. We assume that MLPs are Lipschitz continuous. In our case, $\phi_{\theta_2}$ operates on the sum of $h_{\theta_1}(x)$s for all $N$ number of input points in $A$ (or in $B$). In general, the error made in $h$s may accumulate $N$ times, which causes the precision we must achieve for each individual $h_{\theta_1}(x)$ (compared to $h(x)$) to be $\Theta(\epsilon/N)$. This would have caused the model complexity of $h_{\theta_1}$ to depend on $N$. Fortunately, this is not the case for our encoding

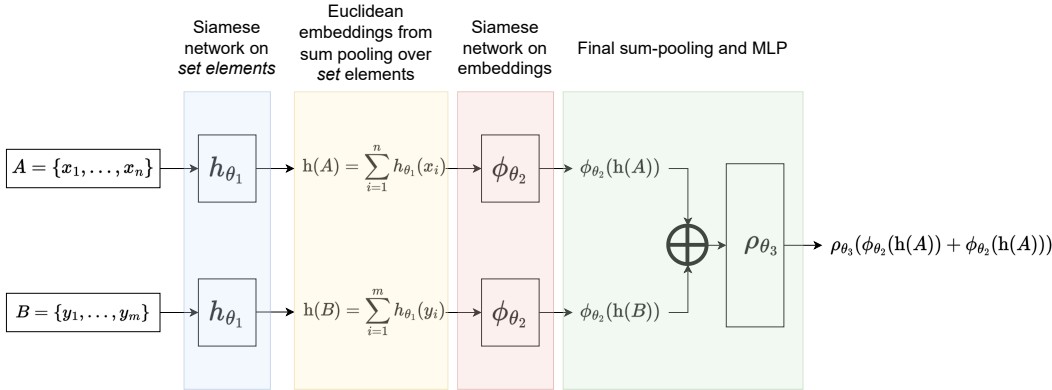

Figure 1: Visually representing a neural network which can universally approximate uniformly continuous SFGI product functions over pairs of point sets.

function $h$. In particular, because our encoding function can be written in the specific form of a normalized sum of individual points in the set $S$; i.e, $\phi_{\theta_2}$ operates on $\sum_{(x,w_x) \in S} w_x h_{\theta_1}(x)$ with $\sum_x w_x = 1$, the error accumulated by the normalized sum will be less than the maximum error from any single point $h(x)$ for $x \in S$. Thus, as both the error for each MLP and the dimension of the domain and co-domain of each approximated function $(\rho, \phi, h)$ do not depend on the size of the input point set $N$, we get that $\rho_{\theta_3}$, $\phi_{\theta_2}$ and $h_{\theta_1}$ each have model complexity independent of the size of the input point set. In short, because the encoding and decoding functions of our sketch can be approximated with neural networks with model complexity *independent* of input point set size, we are able to achieve a low model complexity neural network which can approximate Wasserstein distance arbitrarily well. Note that in general, we are not guaranteed to be able to find a sketch which can be approximated by neural networks which are independent of input size. Finally, this neural network architecture can also be applied to approximating Hausdorff distance. More details regarding Hausdorff distance approximation are available in Appendix A. We summarize the above results in the following corollary.

**Corollary 3.6** *There is a network in the form of $\mathcal{N}_{\mathrm{ProductNet}}$ which can approximate the p-Wasserstein distance between point sets to within an additive $\epsilon$-error. The number of parameters in this network depends on $\epsilon$ and the covering number of $\Omega$ and not on the size of each point set.*

*Additionally, if we replace the sum-pooling with $\max$ in $\mathcal{N}_{\mathrm{ProductNet}}$, there is a network of such a form where we can also approximate Hausdorff distance between point sets to additive $\epsilon$ accuracy.*

**Exponential dependence on dimension** Although the model complexity of $\mathcal{N}_{\mathrm{ProductNet}}$ is independent of the size of the input point set, it depends on the covering number of $\Omega$ which, in turn, can have an exponential dependence on the dimension of $\Omega$. In short, this means that the model complexity of $\mathcal{N}_{\mathrm{ProductNet}}$ has an exponential dependence on the dimension of $\Omega$. However, in practice, many machine learning tasks (e.g. 3D point processing) involve large point sets sampled from low-dimensional space ($d = 3$). Furthermore, in general, the covering number of $\Omega$ will depend on the *intrinsic dimension* of $\Omega$ rather than the *ambient dimension.* For instance, if the input point sets are sampled from a hidden manifold of dimension $d'$ (where $d'$ which is much lower than the ambient dimension $d$), then the covering number would depend only on $d'$ and the curvature bound of the manifold. In many modern machine learning applications, it is often assumed that the data is sampled from a hidden space of low dimension (the manifold hypothesis) although the ambient dimension might be very high.

**Using max-pooling instead of sum-pooling.** Observe that in the final step of combining the Euclidean outputs for two point sets $A, B \in \mathcal{X}$, $\sum_{(x,w_x) \in A} w_x h_{\theta_1}(x) + \sum_{(x,w_x) \in B} w_x h_{\theta_1}(x)$, we use the sum of these two components (as in a DeepSet architecture) : $\sum_{(x,w_x) \in A} w_x h_{\theta_1}(x)$ and $\sum_{(x,w_x) \in B} w_x h_{\theta_1}(x)$, to ensure the symmetric condition of SFGI product functions. One could replace this final sum with a final max such as in PointNet. However, to show that PointNets are able to universally approximate continuous functions $F : K \to \mathbb{R}$ where $K \subseteq \mathbb{R}^{a \times 2}$ is compact, we need

Table 1: Mean relative error between approximations and ground truth Wasserstein distance between point sets. The top row for each dataset shows the approximation quality for point sets with input sizes that were seen at training time; while he bottom row shows the approximation quality for point sets with input sizes that were not seen at training time. Note that $\mathcal{N}_{\mathrm{ProductNet}}$ is our model.

| Dataset | Input size | $\mathcal{N}_{\mathrm{ProductNet}}$ | WPCE | $\mathcal{N}_{\mathrm{SDeepSets}}$ | Sinkhorn |
|---|---|---|---|---|---|
| noisy-sphere-3 | [100, 300] | **0.046 ± 0.043** | 0.341 ± 0.202 | 0.362 ± 0.241 | 0.187 ± 0.232 |
| | [300, 500] | **0.158 ± 0.198** | 0.356 ± 0.286 | 0.608 ± 0.431 | 0.241 ± 0.325 |
| noisy-sphere-6 | [100, 300] | **0.015 ± 0.014** | 0.269 ± 0.285 | 0.291 ± 0.316 | 0.137 ± 0.122 |
| | [300, 500] | **0.049 ± 0.054** | 0.423 ± 0.408 | 0.508 ± 0.473 | 0.198 ± 0.181 |
| uniform | 256 | **0.097 ± 0.073** | 0.120 ± 0.103 | 0.123 ± 0.092 | 0.073 ± 0.009 |
| | [200, 300] | **0.131 ± 0.096** | 1.712 ± 0.869 | 0.917 ± 0.869 | 0.064 ± 0.008 |
| ModelNet-small | [20, 200] | 0.084 ± 0.077 | **0.077 ± 0.075** | 0.105 ± 0.096 | 0.101 ± 0.032 |
| | [300, 500] | **0.111 ± 0.086** | 0.241 ± 0.198 | 0.261 ± 0.245 | 0.193 ± 0.155 |
| ModelNet-large | 2048 | **0.140 ± 0.206** | 0.159 ± 0.141 | 0.166 ± 0.129 | 0.148 ± 0.048 |
| | [1800, 2000] | **0.162 ± 0.228** | 0.392 ± 0.378 | 0.470 ± 0.628 | 0.188 ± 0.088 |
| RNAseq | [20, 200] | **0.012 ± 0.010** | 0.477 ± 0.281 | 0.482 ± 0.291 | 0.040 ± 0.009 |
| | [300, 500] | 0.292 ± 0.041 | 0.583 ± 0.309 | 0.575 ± 0.302 | **0.048 ± 0.006** |

Table 2: Training time and number of epochs needed for convergence for best model

| Dataset | | $\mathcal{N}_{\mathrm{ProductNet}}$ | WPCE | $\mathcal{N}_{\mathrm{SDeepSets}}$ |
|---|---|---|---|---|
| noisy-sphere-3 | Time | 6min | 1hr 46min | 9min |
| | Epochs | 20 | 100 | 100 |
| noisy-sphere-6 | Time | 12min | 4hr 6min | 1hr 38min |
| | Epochs | 20 | 100 | 100 |
| uniform | Time | 7min | 3hr 36min | 1hr 27min |
| | Epochs | 23 | 100 | 100 |
| ModelNet-small | Time | 7min | 1hr 23min | 12 min |
| | Epochs | 20 | 100 | 100 |
| ModelNet-large | Time | 8min | 3hr 5min | 40min |
| | Epochs | 20 | 100 | 100 |
| RNAseq(2k) | Time | 15min | 14hr 26min | 3hr 01min |
| | Epochs | 73 | 100 | 100 |

to use a $\delta$-net for $K$ which will also serve as the intermediate dimension for $\phi$. As $K \subseteq [0, N]^{a \times 2}$ in our case (where $N$ is the maximum cardinality for a point set), the intermediate dimension for a max-pooling invariant architecture at the end (i.e. PointNet) now depends on the maximum size of input point sets.

## 4   Experimental results

We evaluate the accuracy of the 1-Wasserstein distance approximations of our proposed neural network architecture, $\mathcal{N}_{\mathrm{ProductNet}}$, against two different baseline architectures: (1) a Siamese autoencoder known as the Wasserstein Point Cloud Embedding network (WPCE) [15] (previously introduced at the end of Section 1 and is a SOTA method of neural approximation of Wasserstein distance) and (2) a Siamese DeepSets, denoted as $\mathcal{N}_{\mathrm{SDeepSets}}$, which is a single DeepSets model which maps both point sets to a Euclidean space and approximates the 1-Wasserstein distance as the $\ell_2$ norm between output of each point set. As Siamese networks are widely employed for metric learning, $\mathcal{N}_{\mathrm{SDeepSets}}$ model is a natural baseline for comparison again $\mathcal{N}_{\mathrm{ProductNet}}$. We additionally test our neural network approximations against the Sinkhorn distance where the regularization parameter was set to $\epsilon = 0.1$. For each model, we record results for the best model according to a hyperparameter

search with respect to the parameters of each model. Finally, we use the ModelNet40 [31] dataset which consists of point clouds in $\mathbb{R}^3$ and a gene expression dataset (RNAseq) which consists of 4360 cells each represented by 2000 genes (i.e. 4360 points in $\mathbb{R}^{2000}$) as well as three synthetic datasets: (1) uniform, where point sets are in $\mathbb{R}^2$, (2) noisy-sphere-3, where point sets are in $\mathbb{R}^3$, (3) noisy-sphere-6, where point sets are in $\mathbb{R}^6$. The RNAseq dataset is publicly available courtesy of the Allen institute [32]. Additional details and experiments approximating the 2-Wasserstein distance are available in Appendix D.

**Approximating Wasserstein distances.** Our results comparing 1-Wasserstein distance approximations are summarized in Table 1. Additionally, see Table 3 for a summary of time needed for training. For most datasets, $\mathcal{N}_{\text{ProductNet}}$ produces more accurate approximations of Wasserstein distances for both input point sets seen at training time and for input point sets unseen at training time. For the high dimensional RNAseq dataset, our approximation remains accurate in comparison with other methods, including the standard Sinkhorn approximation for input point sets seen at training time. The only exception is ModelNet-small, where the $\mathcal{N}_{\text{ProductNet}}$ approximation error is slightly larger than WPCE for input point set sizes using during training (top row for each dataset in Table 1). However, for point sets where the input sizes were not used during training (bottom row for each dataset in Table 1), $\mathcal{N}_{\text{ProductNet}}$ showed siginificantly lower error than all other methods including WPCE. These results along with a more detailed plot in Figure 2 in Appendix D indicate that $\mathcal{N}_{\text{ProductNet}}$ generalizes better than WPCE to point sets of input sizes unseen at training time. Also, see Appendix D for additional discussion about generalization. Furthermore, one major advantage of $\mathcal{N}_{\text{ProductNet}}$ over WPCE is the dramatically reduced time needed for training (cf. Table 2). This substantial difference in training time is due to WPCE's usage of the Sinkhorn reconstruction loss as the $O(n^2)$ computation time for the Sinkhorn distance can be prohibitively expensive as input point set sizes grow. Thus, our results indicate that, compared to WPCE, $\mathcal{N}_{\text{ProductNet}}$ can reduce training time while still achieving comparable or better quality approximations of Wasserstein distance. Using our $\mathcal{N}_{\text{ProductNet}}$, we can produce high quality approximations of 1-Wasserstein distance while avoiding the extra cost associated with using an autoencoder architecture and Sinkhorn regularization. Finally, all models produce much faster approximations than the Sinkhorn distance (see Tables 3 and 6 in Appendix D). In summary, as compared to WPCE, our model is more accurate in approximating both 1-Wasserstein distance, generalizes better to larger input point set sizes, and is more efficient in terms of training time.

## 5 Concluding Remarks

Our work presents a general neural network framework for approximating SFGI functions which can be combined with geometric sketching ideas into a specific and efficient neural network for approximating $p$-Wasserstein distances. We intend to utilize $\mathcal{N}_{\text{ProductNet}}$ as an **accurate, efficient, and differentiable** approximation for Wasserstein distance in downstream machine learning tasks where Wasserstein distance is employed, such as loss functions for aligning single cell multi-omics data [8] or compressing energy profiles in high energy particle colliders [10, 26]. Beyond Wasserstein distance, we will look to apply our framework to a wide array of geometric problems that can be considered SFGI functions and are desireable to approximate via neural networks. For instance, consider the problems of computing the optimal Wasserstein distance under rigid transformation or the Gromov-Wasserstein distance, which both can be represented as an SFGI function where the factor-wise group invariances include both permutation and rigid transformations. Then our sketch must be invariant to both permutations and orthogonal group actions on the left. It remains to be seen if there is a neural network architecture which can approximate such an SFGI function to within an arbitrary additive $\epsilon$-error where the complexity does not depend on the maximum size of the input set.

## 6 Acknowledgements

The authors thank anonymous reviewers for their helpful comments. Furthermore, the authors thank Rohan Gala and the Allen Institute for generously providing the RNAseq dataset. Finally, Samantha Chen would like to thank Puoya Tabaghi for many helpful discussions about permutation invariant neural networks and Tristan Brugère for his implementation of the Sinkhorn algorithm. This research is supported by National Science Foundation (NSF) grants CCF-2112665 and CCF-2217058 and National Institutes of Health (NIH) grant RF1 MH125317.

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
