# OpenReview forum: "Neural approximation of Wasserstein distance via a universal architecture for symmetric and factorwise group invariant functions"
_NeurIPS.cc/2023/Conference — NeurIPS 2023 poster_

### Official Review · Reviewer_vLSh · 2023-06-21

**Soundness:** 4 excellent
**Presentation:** 3 good
**Contribution:** 3 good
**Rating:** 6
**Confidence:** 5

**Summary:**

The authors propose a framework for learning Wasserstien distances and other `SFGI functions'. Two key ingredients in their approach are the characterization as all SFGI functions (a universality theorem), and a sketching mechanism which shows that the size of certain components in the networks need not depend on the number of points in a set, but only on the dimension of the points and the required approximation error. Empirical verification of the value of this approach is presented.


**Strengths:**

The paper presents two nice ideas:
(a) Characterizaton of SFGI functions (not so suprising in my humble opinion, but still new)
(b) A `sketching' mechanism for the Wasserstein distance which is nice. I think it is worth noting in the paper that the fact that the (psuedo)inverse g of the encoding h can be computed is an advantage over existing encoding which can be proved to be invertible, but we have no good understanding of how to invert them.

Additionally the empirical results seem promising

**Weaknesses:**

 The complexity of the proposed sketching algorithm does not depend on the number of points n, but does depend exponentially on the dimension of the points d. This means that the bounds obtained by the paper would improve upon the bounds obtained by fully injective mappings only for specific values of (d,n,epsilon), namely when epsilon^{-d} is small relative to n. Since in applications often d=3 and n is large, this regime is indeed of interest.


**Questions:**

Suggestions:
Explain a bit more re why to learn Wasserstien distances. Mention you results in the appendix which show better inference time than Sinkhorn.

In Theorem 2.1: the output dimension of phi has considerably been improved recently. The dimension can be 2kn+1 rather than (n+k) choose k. See Corollary 2.2 in [1] and Theorem 3.3 in [2] (full refs below). Presumably this can lead to improved complexity in the subsequent theorem you prove which rely on Theorem 2.1.

Line 127: not sure that the infinity norm is well defined, unless you specify that F_1 and F_2 are subsets of the same C(X,R).

Definition of W_p: D should be replaced by D^p. Also the index of w_n' should be w_m'.

Definition 3.1: specify the group G acts on X.

Lemma 3.2 and elsewhere: do you need to specify that f is uniformly continuous (rather than just continuous) if we are working on compact metric spaces?

Line 185: `then we have that'--> then we can choose

Line 191: not clear why the assumption of a topological embedding is strong, given Theorem 2.1. Is it because injectivity is not the same as a topological embedding? Or maybe the issue is the dimensionality?

Line 208: i.e., MLP should be e.g. MLP

Line 223 and Theorem 3.4: I would consider just saying W_p is the distance without introducing d_{fancy X} which is confusing since there is also d_X

I would view the construction in Theorem 3.4 as a smoothened version of one-hot encoding. You might want to mention this connection.

Line 238: M was N previously, better to be consistent.

Line 244: I'm not sure whether d_max was defined? (presumably it is the diameter of X)

The first inequality just under 254 could be explained more: you're basically saying that some matrix is a transport plan between the measures defined by S and g(h(S)).

The second line in said inequality has an explanation in the middle, this could be styled better.

In Corollary 3.5: you do not specify that delta depends on epsilon. The quantifiers should be reoredered. Also note that for the f you are interested in, namely the W_p distance, f is actually 1-Lipschitz so you can take delta=epsilon.

Corollary 3.6: mention dependence on dimension not only on epsilon.

Line 354: wasn't completely clear whether this was referring to the permutation+orthogonal problem from the previous line or this was a new problem which I didn't fully understand



[1] Low Dimensional Invariant Embeddings for Universal Geometric
Learning, https://arxiv.org/pdf/2205.02956.pdf
[2] Neural Injective Functions for Multisets, Measures
and Graphs via a Finite Witness Theorem https://arxiv.org/pdf/2306.06529.pdf

**Limitations:**

Exponential dependence on dimension of points in the set should be mentioned.

---

> ### Author Rebuttal · Authors · 2023-08-09
>
>
> Thank you for your very helpful comments! We will accommodate those in our revised manuscript (including mentioning explicitly the dependence on (the intrinsic) dimension). Below we will just mention a couple more major ones.
>
> - Regarding your comment on **exponential dependence on dimension of input points**: Indeed, that is the case. However, note that in general, this is the ''intrinsic dimension" of the domain where points are sampled from, instead of the ``ambient dimension". For example, if the input points are sampled from a hidden manifold of dimension $m$, then the covering number will depend only on $m$ and the curvature bound of the manifold. One could also get a bound on a collection of manifold pieces. In other words, it really depends on the intrinsic dimension of the hidden domain points are sampled from, although for simplicity, we use the Euclidean dimension in our paper. We also note that in modern machine learning, we often assume (explicitly or implicitly) that data is sampled from a hidden space of low dimensions (e.g., in the manifold hypothesis), even though the ambient dimension might be very high. We will make this point more clear and explicit in the revised manuscript.
>
> - Thank you for your references regarding improved bounds for (multi)set functions. We will include them into the revised manuscript. Indeed, using results from the provided references, the dimension of the latent space in Lemma 3.3 becomes $2\cdot a \cdot m + 1$ and the dimension of the latent space in Corollary 3.4 becomes $2\cdot (a(\delta)) \cdot 2 + 1$.
>
> - Regarding **Line 191**, what we meant is that a topological embedding requires injectivity, while in a sketch, one can collapse input objects as long as after decoding, we obtain an approximated object close to input.

---

> > ### Comment · Reviewer_vLSh · 2023-08-11
> >
> > I am happy with the authors' rebuttal. Will retain my rating of 6. Thanks

---

### Official Review · Reviewer_xMX9 · 2023-07-04

**Soundness:** 3 good
**Presentation:** 3 good
**Contribution:** 3 good
**Rating:** 5
**Confidence:** 4

**Summary:**

The paper introduces the concept of symmetric and factor-wise group invariant functions (SFGI), which are continuous and symmetric product functions on complex objects like point sets and graphs. The authors propose a general neural network architecture for approximating SFGI functions and combine it with a sketching idea to develop a specific and efficient neural network that approximates the p-th Wasserstein distance between point sets. The key contribution is that the model complexity of the proposed neural network is independent of the input point set sizes, which is a novel result in the field. The paper demonstrates the effectiveness of the neural network architecture through empirical evaluations, comparing it with other models including a state-of-the-art Siamese Autoencoder. The proposed architecture outperforms existing models in terms of generalization and training speed. The authors highlight the potential application of their framework in solving various geometric optimization problems.

**Strengths:**

1. *Novel Contribution*: The paper introduces the concept of symmetric and factor-wise group invariant functions (SFGI) and presents a general neural network architecture for approximating these functions. This contribution addresses the need for distance functions on complex objects that are invariant to various group actions, such as permutation or rigid transformation.

2. *Bounded Model Complexity*: The authors demonstrate that their proposed neural network architecture for approximating SFGI functions has a bounded model complexity. This result is significant because it shows that there exists a neural network with the capacity to approximate the p-th Wasserstein distance without the complexity depending on the sizes of the input point sets.

3. *Integration of Sketching Ideas*: The paper integrates sketching ideas with the general neural network architecture to develop an efficient and specific neural network for approximating the p-Wasserstein distance between point sets. This integration adds a practical element to the theoretical framework and contributes to the overall efficiency of the approach.

4. *Potential Applications*: The authors discuss potential applications of their framework beyond Wasserstein distance estimation. They highlight its potential use in solving a broad range of geometric optimization problems, such as k-means in a metric space.


**Weaknesses:**

1. *Limited Comparison*: While the paper compares the proposed neural network architecture with a state-of-the-art Siamese Autoencoder and other models, the comparison may not be comprehensive enough. Authors consider only the evaluation of the accuracy of the 1-Wasserstein distance. It would be beneficial to include also case p=2  to better understand the relative strengths and weaknesses of the proposed approach.

2. *Limited Generalizability*: While the proposed framework shows promising results for approximating the p-Wasserstein distance between point sets, the paper does not extensively explore its generalizability to other types of complex objects or distance functions. Further investigation into the applicability of the framework to a broader range of geometric matching problems would strengthen the overall contribution.

**Questions:**

See Section 'Weaknesses'.

**Limitations:**

The authors should devote more space to the limitations of the proposed approach.

---

> ### Author Rebuttal · Authors · 2023-08-09
>
>
> Thank you for your comments!
>
> Regarding your comments on **limited comparison**: Thank you, and we have now also carried out experiments w.r.t. 2-Wasserstein distance. See our results in the attached PDF. As we reported at the beginning in the general comments to all reviewers, we can see that the improvements over all other ML methods in terms of both accuracy and speed are very similar to the 1-Wasserstein distance. Note that for the Sinkhorn distance, there is a tradeoff of time complexity versus accuracy via the regularization parameter $\epsilon$. We choose a sufficiently small $\epsilon$ so that Sinkhorn has high accurary as shown in Table 1 ($\epsilon = 0.01$). However, as we show in Table 2, the speed is much more slower than ours. In fact, if we increase the regularization parameter $\epsilon$ to $0.10$, Sinkhorn is still slower than ours even on small dataset, yet the accuracy is already much worse than our neural approximation $\mathcal{N}_{\mathrm{ProductNet}}$. On average, the Sinkhorn with respect to $\epsilon = 0.01$ is 20 times slower than our method on datasets with small input point set size (100-300) and 80 times slower than our method on datasets with larger input point set size (400-600). Note that the gap is increasing as the Sinkhorn distance takes quadratic time to compute.
>
> To further address your point regarding limited comparison, we also show the effectiveness of our method for higher dimensional data. Here we take a single cell gene expression data set $P$ consisting of $4441$ cell each represented by $4000$ genes (i.e., $4441$ points in $\mathbb{R}^{4000}$ space. We subsample 3000 pairs of point sets of size 20 to 200 for training, and the testing set consists of 300 pairs of point sets of size 20 to 200. As we can see, our neural estimation remains accurate (see the comparison with Sinkhorn). Unfortunately, due to constraints on computational resources and time, we were not able to generate comparisons to other ML methods (WPCE, $\mathcal{N}_{\mathrm{SDeepSets}}$) for RNAseq.
> See the attached PDF for the additional experimental results, and we will add these results to the revised manuscript.
>
> Regarding your comments on **limited generalizability**: First, we want to point out that just having an efficient and effective neural model to estimate the Wasserstein distance itself is of great interests, which have already attracted various past work (in addition to what we have in Related work of our manuscript, there are also several very recent pieces of work emerging).
>
> Our response to Reviewer CxcX is also relevant, which for completeness we include here as well:
>
> In the appendix, we showed that the same framework can work for Hausdorff distance (but with max pooling, similar to PointNet). In general, we think that we might be able to extend these to geometric matching such as Frechet distance for curves (and variance of Frechet distance) without much change of the architecture. A more challenging next step is to compute optimal Wasserstein distance between point-sets under rigid transformations: While there are ways to handle rotation invariance (basis invariance, e.g., via the approach of [Lim et al., ICLR 2023]), how to do so with bounded model complexity (for the encoder) is still challenging; but we think this is achievable.
> Developing such a neural estimation for graph distances is also very interesting -- with suitable distances for the space of graphs and proper assumptions, one might be able to show the existence of low dimensional sketch, but how to argue that the encoder needed has bounded model complexity is more challenging.
>
> Finally, while it is not stated in the paper, our current framework can be directly extended to compute the Frechet variance, or, the average distance to 1-median of a set of $k$ pointsets $P_1, \ldots, P_k$ (see the response to Reviewer 1U84 for the definition of Frechet mean/Frechet variance, or 1-median and average ditsance to it). Roughly speaking, the Frechet mean of a set of point-sets can be thought of as their ``center'' minimizing the Frechet variance (which is the total squared distance from the mean-pointset to each input pointset).
>
> We will add more discussions in the revised manuscript.

---

### Official Review · Reviewer_CxcX · 2023-07-07

**Soundness:** 4 excellent
**Presentation:** 4 excellent
**Contribution:** 4 excellent
**Rating:** 6
**Confidence:** 4

**Summary:**

The goal of this submission is to learn the (Wasserstein) distance between complex objects (e.g. point sets, graphs) within an arbitrary additive $\epsilon$-error.

This paper presents a general neural network architecture for approximating symmetric and factor-wise group invariant (SFGI) functions.

The proposed SFGI neural network could achieve universality with the number of parameters independent of the input point set size.

The experiment results show the proposed SFGI neural network could approximate the (best or comparable) Wasserstein distance between point sets.

**Strengths:**

This paper presents a general neural network framework for approximating the Wasserstein distance between point sets.

The size of the proposed universal neural network approximator is independent w.r.t. the input size, which is new and important. The above advantage is demonstrated in Table 2 in the main paper.

The proposed neural network approximator is simple but may inspire dozens of downstream network designs.

The presentation is clear. Well written.

The code is provided.

I haven't checked the supplementary carefully.

**Weaknesses:**

The proposed neural uses sum-pooling, so that the parameter size will be independent of the input point set size. I am looking forward to seeing more detailed discussions of it. e.g. How could the proposed neural deal with varying input point set size (which is common in many applications)?


**Questions:**

in table 1, the row of 'uniform', it seems Sinkhorn achieved a better Wasserstein distance estimation rather then the proposed model, so the wrong highlight.

Any insights for different distances and different complex objects (e.g. graph)? Should the proposed model to change accordingly?

---

> ### Author Rebuttal · Authors · 2023-08-09
>
>
> Thank you for your comments!
>
> Regarding your question on the handling of **points of varying sizes** (from your comment ``I am looking forward to seeing more detailed discussions of it ...''): First, in our theoretical results, the fact that the parameter size (model complexity) is independent of input point size essentially is from the constructive proof of Theorem 3.4. The proof shows not only that a latent space of dimension independent to $n$ exists, but also that this can be achieved by a bounded set of functions $h_i$s (forming the encoder), for $i\in [1, a]$, each of which has a simple form and will later be replaced by a small MLP of constant size in our final neural model $\mathcal{N}_{ProductNet}$.
> Note that a normalization is happening as the input is a weighted point set with total weight $1$ -- one can think that if there are $n$ input points in a point-set, each point receives weight $1/n$.
>
> While we assume a maximum size of input pointsets for simplicity in formulating and stating our results, our final neural model can handle any input pointset size (similar to DeepSet, PointNet, or message passing GNNs) -- indeed, in our experiments, we often train on point-sets of small sizes (100 to 300 points) and show that the final model generalizes very well to point sets of much larger sizes unseen during the training (see the second row for each dataset in Table 1): we tested up to around 2000 points -- as the accuracy degrades only very slightly with larger point sets, we expect that it perform similarly for much larger sizes as well.
>
> We will make these points more clear in our revised manuscript.
>
> Regarding ''Any insights for different distances and different complex objects (e.g., graphs) ...'': That is a very good question. In the appendix, we showed that the same framework can work for Hausdorff distance (but with max pooling, similar to PointNet). In general, we think that we might be able to extend these to geometric matching such as Fréchet distance for curves (and variance of Fréchet distance) without much change of the architecture. A more challenging next step is to compute optimal Wasserstein distance between point-sets under rigid transformations: While there are ways to handle rotation invariance (basis invariance, e.g., via the approach of [Lim et al., ICLR 2023]), how to do so with bounded model complexity (for the encoder) is still challenging; but we think this is achievable.
> Developing such a neural estimation for graph distances is also very interesting -- with suitable distances for the space of graphs and proper assumptions, one might be able to show the existence of low dimensional sketch. However, arguing that the encoder needed has bounded model complexity is more challenging.
>
> Finally, while it is not stated in the paper, our current framework can be directly extended to compute the Fréchet variance or the average distance to 1-median of a set of $k$ pointsets $P_1, \ldots, P_k$ (see the response to Reviewer 1U84 for the definition of Fréchet mean/Fréchet variance, or 1-median and average distance to it). Roughly speaking, the Fréchet mean of a set of point-sets can be thought of as their ''center'' minimizing the Fréchet variance (which is the total squared distance from the mean-pointset to each input pointset).
>
> We will add more discussions in the revised manuscript.

---

### Official Review · Reviewer_1U84 · 2023-07-07

**Soundness:** 3 good
**Presentation:** 3 good
**Contribution:** 3 good
**Rating:** 5
**Confidence:** 4

**Summary:**

The paper proposes a neural network architecture that efficiently approximates $p$-Wasserstein distance between point sets. This is achieved by exploiting the networks' capability to approximate functions that are symmetric and invariant to group actions componentwise. The highlight of the model is that its complexity remains independent of the input sample sizes, improving generalization capabilities. Supporting empirical evidence also showcases the time efficiency of the approach compared to SOTA architectures.

**Strengths:**

The paper is well-written, with ample theoretical discussion regarding the problem and sound proofs. The empirical evidence provided supports the theoretical claims to a fair extent.

**Weaknesses:**

The dimension of the range of functions, used to approximate uniformly continuous and permutation invariant maps $f(A, B)$, turn out to be independent of the size of input points, rather depend on the covering number of the input space. Under the assumption of compactness, this becomes straightforward as a finite cover is ensured. As such, the real challenge lies in showing the result under unbounded domains, perhaps with decaying tail conditions (e.g. sub-Exponential) on input distributions.

**Questions:**

1. What is meant by '$1$-median' of a collection of point sets? [Line 39] Also, specify the notation $S$ in Definition 3.1.

2. Shouldn't it be $D_{i,j}=d_{X}(x_{i},x'_{j})^{p}$ instead in Line 153?

3. In dimensionality reduction or embedding problems it is often difficult to find out continuous encoding maps. For example, the very problem of Autoencoders is mostly dedicated to ensuring that $d_{\mathcal{X}}(g \circ h(\cdot), \cdot) <\delta$. In that light, the assumption that sketches readily exist seems quite strong (as in Lemma 3.2). The paper itself expresses the concern in Line 287. Perhaps the authors can provide some examples to justify. This seems crucial since in most results regarding embedding onto Euclidean spaces (due to Johnson-Lindenstrauss, Bourgain, etc.) the optimal embedding dimension under a given level of distortion depends on the input sample size (cardinality of the finite input space).

4. The Wasserstein distance operates as a metric on the class of probability distributions defined on $X$ (say, $\mathcal{P}(X)$). So, is it perhaps more appropriate to define $\mathcal{X}$ as the set of weighted Dirac deltas corresponding to points from $X$?


**Limitations:**

The paper has no immediate negative societal impact.

---

> ### Author Rebuttal · Authors · 2023-08-09
>
> Thank you for your comments!
>
> Regarding your main comment that "Under the assumption of compactness, this becomes straightforward as a finite cover is ensured.", we would like to clarify the following:
> Note that there are **two spaces** involved in our problem. The first space $\Omega \subset (X, d_X)$ is the metric space from which input point sets are sampled -- in particular: we take $X = \mathbb{R}^d$, metric $\mathrm{d}_X$ to be the standard Euclidean distance $\| \cdot \|_2$, and $\Omega$ to be a compact set in $\mathbb{R}^d$.
>
> The second space, denoted by $(\mathcal{X}, d_{\mathcal{X}})$ is the space of weighted point-sets (of maximum cardinality $N$, where points are from $\Omega \subset (X, d_X)$). This space, $(\mathcal{X}, d_{\mathcal{X}})$, is a factor of the metric space where our product function $f(A, B)$ is defined on; that is, $f: \mathcal{X} \times \mathcal{X} \to \mathbb{R}$. In our setting, if input points are from $\Omega \subset \mathbb{R}^d$, then $\mathcal{X} = \Omega^N \subset \mathbb{R}^{N d}$, and note that the metric $d_{\mathcal{X}} = W_p$, the $p$-th Wasserstein distance for point-sets.
>
> While $\mathcal{X}$ is compact when $\Omega$ is compact, note that its dimension is $Nd$, which depends on the maximum size of input point-set. We can not directly use a bound of covering number of this space $\mathcal{X}$ (which will naively depend on the dimension of $\mathcal{X}$ which is $Nd$) to define a model as we want the number of parameters to be *independent* of the maximum input point set size $N$.Instead, our Theorem 3.4 shows that when the metric for $\mathcal{X}$ is $d_{\mathcal{X}} = W_p$ (the $p$-th Wasserstein distance), we can find a sketch whose latent dimension depends on the covering number of $\Omega$.
> Our notations of $(X, d_X)$ and $(\mathcal{X}, d_{\mathcal{X}})$ might have caused confusion. We will update them to $(\Omega, \ell_2)$ and $(\mathcal{X}, W_p)$ in the revised manuscript to make the distinction more clear.
>
> Furthermore, the existence of a low-dimensional latent space **does not** necessarily imply that one can approximate the map from the original domain to this latent space via a model of bounded complexity.
> To this end, note that our proof in Theorem 3.4 is constructive: it has specific form using a bounded number of functions, $h_i$'s (for $i\in [1, a]$), each of which has a simple closed form and can be replaced by a MLP of complexity independent to input point set size.
>
> Regarding "What is meant by '1-median' of a collection of point sets?":
> Given a collection of $k$ objects $\{ Y_1, \ldots, Y_k\}$ sampled from a metric space $(\mathcal{Y}, d_{\mathcal{Y}})$, its {\it 1-median}, or geometric median, is defined as
> $$ Y^* = argmin_{Y} \sum_{i=1}^k d (Y, Y_k),$$
> which intuitively, is the center of these $k$ objects. For example, if each $Y_i$ is a point set, one can think of $Y^*$ as its average point-set. In our sentence (line 39), we in fact refer to the total distance to the 1-median (or one can think of this as the cost of 1-median), namely,
> $$ cost(Y_1, \ldots, Y_k) = \min_Y \sum_{i=1}^k d (Y, Y_k) = \sum_{i=1}^k d (Y^*, Y_k).$$
> If we use squared distance $d^2 (Y, Y_k)$ in the above definitions, then we obtain 1-mean (sometimes also called the Fréchet mean) and its corresponding cost is Fréchet variance. Both 1-median (and its associated cost) and the 1-mean (and the associated Fréchet variance) are commonly used in statistical analysis of collections of objects $Y_i$s (e.g., computing mean / variance in the space of pointsets). While we didn't explicitly state it in the paper, our results / architectures also hold for computing the cost of 1-median or Fréchet mean of a fixed set of pointsets sampled from a compact set in $\mathbb{R}^d$. In general, there is no closed form solution for 1-median or 1-mean for complex spaces, and they are often computed empirically by solving a non-convex optimization procedure.
>
> Regarding your comment "In dimensionality reduction or embedding problems ...": We note that having a sketch for space of point-sets (under Wasserstein distance or some other distance) is different from standard dimensionality reduction or embedding, which often aims to preserves distance metric in the latent space. In our case, we only need the latent space to provide sufficient information so that the Wasserstein distance can be approximately **after decoding**. There is no distance preservation in the latent space.
> At least for the case of Wasserstein distance, it turns out that it is sufficient to achieve this by mapping to $\mathbb{R}^p$ where $p$ depends only on the covering number of the input domain. A low covering number (and by extension, a lower dimension for the latent space $\mathbb{R}^p$) can be obtained if the domain $\Omega$ has low ``intrinsic dimension'', such as a low-dimensional Euclidean space $\mathbb{R}^d$ in the linear case, or a hidden non-linear manifold with low intrinsic dimension $d$ and bounded curvature (and the ambient dimension could be high), or a collection of constant number of manifold pieces.
> In our paper, we use compact subsets of Euclidean space $\mathbb{R}^d$ for simplicity, but our results can be extended to points sampled from a hidden manifold of dimension $m$ with bounded curvature -- in which case the covering number is also bounded (e.g., using results of [Roër 2013]).
>
> We note that the existence of a low-dimensional hidden structure in data is an important assumption for modern machine learning in many settings. For instance, while the space of natural images has very high dimension (size of an image), the intrinsic dimension of the hidden space where real images lie is far smaller.
>
> Thank you for your suggestion of defining $\mathcal{X}$ as weighted Dirac measures. We will incorporate that in the revision.

---

> > ### Comment · Reviewer_1U84 · 2023-08-18
> > **Response to the rebuttal**
> >
> > I appreciate the detailed response by the authors. Taking into account the other reviews and corresponding rebuttals, I stick to my initial evaluation.

---

### Author Rebuttal · Authors · 2023-08-09

We thank all reviewers for their valuable comments and feedback.

We are happy that reviewers appreciate our contribution of a simple neural network model (constructed based on our theoretical insights) which can estimate Wasserstein distance accurately with bounded model complexity.
Indeed, identifying the right neural model for a given problem, especially for an optimization problem, is not often an easy task and the role of representation learning is crucial, especially for complex objects (point-sets in our case).
At a high level, we are interested in problems relating to geometric matching of complex objects, and we focus on the Wasserstein distance between point-sets in this paper.
Our theoretical analysis implies that a simple neural model consisting of a Siamese architecture **followed by MLP layers at the end (which is crucial)** turns out to be universal with respect to the Wasserstein distance problem and the model complexity is independent of input point sets (see Corollary 3.6 and Figure 1).
Empirically, we emphasize that our neural model outperforms both Sinkhorn and SOTA ML-based approaches for Wasserstein distance computation in terms of **both accuracy and speed**, and also generalizes significantly better than SOTA ML-based approaches.

To further demonstrate the effectiveness of our model, as suggested by Reviewer xMX9, we also:

(1) carry out experiments evaluating the accuracy of our approximation for $W_2$, i.e., $p$-th Wasserstein distance for $p = 2$; and

(2) further tested our accuracy on a high dimensional real RNAseq dataset (in 4000 dimensions), which are gene expression data.


See the attached pdf for results.
For the case of 2-Wasserstein distance, Table 1 in the pdf shows a similar trend the 1-Wasserstein distance originally reported in our paper. In fact, the improvement in terms of accuracy over other ML models are even bigger.
Note that for the Sinkhorn distance, there is a tradeoff of time complexity versus accuracy via the regularization parameter $\epsilon$. We choose a sufficiently small $\epsilon$ so that Sinkhorn has high accurary as shown in Table 1. However, as we show in Table 2, the speed is much more slower than ours. In fact, if we increase the regularization parameter $\epsilon$ to $0.10$, Sinkhorn is still slower than ours even on small dataset, yet the accuracy is already much worse than our neural approximation $\mathcal{N}_{ProductNet}$. On average, the Sinkhorn with respect to $\epsilon = 0.01$ is 20 times slower than our method on datasets with small input point set size (100-300) and 80 times slower than our method on datasets with larger input point set size (400-600). Note that the gap is increasing as the Sinkhorn distance takes quadratic time to compute.

We also point out that a neural approximation of Wasserstein distance itself, or more generally, optimal transport, is interesting on its own and has already attracted extensive past work (see our related work section), as well as recent new work.
In our Supplementary material, we show that our neural model can be easily extended to Hausdorff distance. Interesting future work will include extending the model to estimate Fréchet distance (or variants) for curves, or optimal Wasserstein / Hausdorff distance under rigid transformations.

---

### Decision · Program_Chairs · 2023-09-21

**Decision:**

Accept (poster)

**Comment:**

In this paper the authors propose a novel way to approximate Wasserstein distance with neural networks. The reviewers found the paper interesting but had some concerns about the assumptions, the dependence of the method to input size and about the experiments.

The authors did a detailed rebuttal to those concerns with new experiments, and clarifications that was appreciated by reviewers. All reviewers kept their borderline positive score and are OK for accepting the paper that brings a novel architecture for efficient approximation of the Wasserstein distance to the community.

The authors must take into account the reviewers comments and integrate the results and discussions (exponential dependence of the dimension of the space for instance) from the rebuttal in the final version of the paper.